# Parasomnias in Pregnancy

**DOI:** 10.3390/brainsci13020357

**Published:** 2023-02-18

**Authors:** Jitka Bušková, Eva Miletínová, Radana Králová, Tereza Dvořáková, Adéla Tefr Faridová, Hynek Heřman, Kristýna Hrdličková, Antonín Šebela

**Affiliations:** 1National Institute of Mental Health, Topolová 748, 250 67 Klecany, Czech Republic; 2Third Faculty of Medicine, Charles University, 100 00 Prague, Czech Republic; 3The Institute for the Care of Mother and Child, 147 10 Prague, Czech Republic; 4Second Faculty of Medicine, Charles University, 150 06 Prague, Czech Republic; 5Faculty of Arts, Charles University, 116 38 Prague, Czech Republic

**Keywords:** pregnancy, NREM parasomnias, REM parasomnias, headache

## Abstract

Objectives: Pregnancy is often associated with reduced sleep quality and an increase in sleep disorders, such as restless leg syndrome, obstructive sleep apnea, and insomnia. There are few studies investigating the prevalence of parasomnias in pregnancy, although they may be expected to be a significant problem, as disturbed sleep in this time period in addition to these sleep disorders may trigger parasomnia episodes. Methods: We conducted a survey using an online questionnaire focusing on a comparison of the prevalence of parasomnias in three time periods: 3 months before pregnancy, during pregnancy, and 3 months after delivery. We also inquired about psychiatric and neurological comorbidities, current anxiety and depression symptoms, and pregnancy complications. Results: A total of 325 women (mean age 30.3 ± 5.3 years) participated in the online survey. The overall number of reported parasomnias increased during pregnancy compared to the 3 months before pregnancy (*p* < 0.001) and decreased after childbirth (*p* < 0.001). Specifically, we found a significant increase in sleepwalking (*p* = 0.02) and night terrors (*p* < 0.001), as well as in vivid dreams (*p* < 0.001) and nightmares (*p* < 0.001) during pregnancy. A similar significant increase during pregnancy was reported for head explosion (*p* < 0.011). In contrast, the number of episodes of sleep paralysis increased after delivery (*p* = 0.008). At the individual level, an increase in the severity/frequency of individual parasomnia episodes was also observed during pregnancy. Participants whose vivid dreams/nightmares persisted after delivery had higher BDI-II and STAI-T scores. Our data also suggest a significant impact of migraines and other chronic pain, as well as complications during pregnancy, on the presence of parasomnia episodes in our cohort. Conclusions: We have shown that the prevalence of parasomnias increases during pregnancy and needs to be targeted, especially by non-pharmacological approaches. At the same time, it is necessary to inquire about psychiatric and neurological comorbidities and keep in mind that more sleep disorders may be experienced by mothers who have medical complications during pregnancy.

## 1. Introduction

Pregnancy is a physiological state associated with numerous hormonal, physical, and behavioral changes that may themselves cause significant alterations in the quality and duration of sleep [1,2,3]. In addition, pregnant women are also affected by an increased rate of sleep disorders such as insomnia [4], sleep-disordered breathing (SDB) [5], and restless legs syndrome (RLS) [6,7]. As a result, it is not surprising that a majority of pregnant women report sleep disruption, which is known to have significant health implications and pose an increased risk of adverse pregnancy outcomes [8,9,10].

Although many sleep disorders in pregnancy are well described, very little is known about the prevalence of parasomnias during pregnancy, and the results of the sporadic studies on this topic are not consistent [11,12,13]. Parasomnias represent undesirable events or behavior occurring at sleep onset or during sleep, which disturb the patient or others, and can even lead to unconscious violence in sleep [14,15]. Disturbed sleep during pregnancy, as well as the aforementioned sleep disorders such as sleep-disordered breathing or movement disorders [16], may constitute significant triggers for parasomnia episodes and increase their occurrence during pregnancy.

Therefore, the main aim of this study was to determine the prevalence and severity of different types of parasomnias in pregnancy, and to compare their occurrence in pregnancy with the pre-pregnancy and postpartum periods. We were also interested in whether their occurrence is related to current anxiety and depressive symptoms, psychiatric and neurologic comorbidities, or complications during pregnancy.

## 2. Materials and Methods

Through a questionnaire study, we reached out to mothers who had given birth between 3 and 36 months prior to taking part in the survey. The online survey targeting the community of pregnant or new mothers was distributed through social networks between August and October 2022. Information and posters publicizing the study were displayed on the social media profiles of the National Institute of Mental Health, and the study recruitment materials were published on the profiles of Úsměv mámy, a major Czech patient organization for peri- and postnatal mental health. Furthermore, using the snowball sampling method, other institutions and individuals were involved in the recruitment process. A web link to the data collection form and informed consent for study participation were a part of the study recruitment materials. Along with an informed consent section, the data collection form included the following items: age, number of births and abortions, complications during pregnancy, psychiatric and neurological comorbidities, and self-reported questions to determine the prevalence of each parasomnia (Table 1) in three time periods: (1) in 3 months before pregnancy; (2) during pregnancy; and (3) in 3 months after delivery. Additionally, the respondents were asked to fill in the Beck Depression Inventory (BDI-II) [17] and the State-Trait Anxiety Inventory (STAI-S and STAI-T) questionnaires [18]. To quantify the responses, we categorized the occurrence of parasomnias in each time period as I/never or less than once per month; II/less than once per week; III/1-2 nights per week; IV/5 nights per week; and V/every night or almost every night. Participants who did not answer all the questions were excluded from the analysis.

### Statistical Analysis

The data were checked for normality of distribution using the Shapiro–Wilk test. Subsequently, the frequencies of each response were calculated and general scores were applied. Then, the frequencies of each parasomnia in a particular phase were compared using the McNemar–Bowker test of symmetry. In order to investigate the relationship between psychiatric symptoms and parasomnias, we used bivariate correlations and calculated Spearman’s rho. The difference between the reported frequencies of particular symptoms among participants who reported neurological, psychiatric, or other complications during pregnancy and those who did not was measured by an analysis of variance (ANOVA) for repeated measures. The same test was used to compare the severity of particular symptoms across the three time periods, and to test the difference in severity of these symptoms in patients taking antidepressants (more specifically, selective serotonin reuptake inhibitors). The level of significance was set at <0.05 for all the statistical tests.

The study was approved by the local ethics committee of the National Institute of Mental Health, Czech Republic.

## 3. Results

A total of 977 respondents participated in the questionnaire study, but only 325 (33.3%) of them completed all the required questions. Participants who did not complete the questionnaire were excluded from further analysis without identifying their reasons for non-completion. The mean age was 30.3 ± 5.3 years; 260 (80%) women were primiparas and 65 women (20%) were multiparas. Abortion was reported in a total of 62 (19.1%) cases. Complications during pregnancy were indicated by 72 (22.2%) participants. Psychiatric comorbidities were present in 18 (5.5%) cases and neurological comorbidities in 16 (4.9%) cases.

Any parasomnia was reported by 213 (65.6%) of the respondents in the total evaluated sample: before pregnancy in 67 (20.5%), during pregnancy in 81 (24.8%) and after pregnancy in 66 (20.3%) cases. The overall number of reported parasomnias increased during pregnancy compared to the 3 months before pregnancy (*p* < 0.001) and decreased after childbirth (*p* < 0.001). The representation of the individual parasomnias is shown in Figure 1 (the number of those who had a defined parasomnia during a specific time period). Parasomnias were present simultaneously at all three time periods in the following numbers of participants: sleeptalking, n = 36; screaming, n = 8; confusional arousal, n = 45; sleepwalking, n = 3; sleep terrors, n = 28; sleep-related eating disorder, n = 4; sexsomnia, n = 9; sleep paralysis, n = 20; hypnagogic hallucinations, n = 45; vivid dreams, n = 159; nightmares, n = 123; aggressive dreams, n = 65; dream-related behavior, n = 9; head explosion, n = 25; violence in sleep, n = 5.

We found a significant increase in vivid dreams (*p* < 0.001) and nightmares (*p* < 0.001) as well as sleepwalking (*p* = 0.02) and night terrors (*p* < 0.001) during pregnancy in comparison to the time period before pregnancy, and these symptoms generally persisted 3 months after delivery (*p*< 0.001). A similar significant increase during pregnancy was reported for head explosion (*p* < 0.011). In contrast, the number of episodes of sleep paralysis increased after delivery (*p* = 0.008). The incidence of other types of parasomnias did not significantly change during pregnancy and in the postpartum period. Further data are presented in Table 2. The table shows the number of participants who suffered from a defined parasomnia at a given frequency in a specified time period.

We further assessed how the severity of parasomnias varied across the three time periods within individuals. The severity of some parasomnia symptoms increased during pregnancy compared to the three months before pregnancy. This was true for confusional arousal (*p* = 0.006), sleep terrors (*p* < 0.001), sleep-related eating disorder (*p* = 0.026), vivid dreams (*p* < 0.001), nightmares (*p* < 0.001), aggressive dreams (*p* < 0.001), and head explosion (*p* = 0.001). In contrast, the severity of some of these symptoms decreased in the three months after delivery compared to the time during pregnancy. This occurred in sleeptalking (*p* = 0.005), sexsomnia (*p* = 0.011), sleep paralysis (*p* = 0.002), confusional arousal (*p* = 0.013), vivid dreams (*p* < 0.001), nightmares (*p* < 0.001), aggressive dreams (*p* = 0.002), and head explosion (*p* = 0.036). Some of the symptoms remain more severe three months after delivery compared to three months before pregnancy—specifically, confusional arousal (*p* < 0.001), sleep terrors (*p* = 0.001), and vivid dreams (*p* < 0.001).

We also found a moderate positive correlation between the occurrence of nightmares and BDI-II during 3 months after delivery (Spearman’s rho = 0.219, *p* < 0.001), and a weak positive correlation between the occurrence of vivid dreams and BDI-II (Spearman’s rho = 0.187, *p* = 0.001) and STAI-T (Spearman’s rho = 0.139, *p* = 0.012) scores during the same time period after delivery. Psychiatric comorbidities were associated with vivid dreams (*p* = 0.046) and hypnagogic hallucinations (*p* = 0.049) during all the observed time periods. A total of 14 participants were taking SSRI (selective serotonin reuptake inhibitors), all in monotherapy (sertraline in seven cases, citalopram in four cases, escitalopram in two participants, and fluvoxamine in one case). Generally, we found no significant differences in the occurrence of parasomnias related to the administration of antidepressants. Most participants reported the same frequency of episodes in all three follow-up periods, with only three subjects showing an increase in the number of episodes during pregnancy (these participants were on sertraline, escitalopram, and fluvoxamine monotherapy). Similarly, we observed an increase in dream-related behavior during pregnancy in only two subjects (on citalopram and fluvoxamine monotherapy). Younger women generally demonstrated higher scores in both STAI-T and STAI-S (*p* = 0.021; *p* < 0.001).

Our data also suggest a significant impact of existing neurological disorders on the presence of parasomnia symptoms. Patients most commonly reported migraines (with/without aura; eight cases), epilepsy (one case), tension-type headaches (one case), and chronic vertebral pain (two cases). Combined symptoms were found in four cases. We found a statistically significant effect of existing neurological disorders on the occurrence of night screaming (*p* = 0.047), sleepwalking (*p* < 0.001), sleep-related eating disorders (*p* < 0.001), sleep paralysis and hypnagogic hallucinations (both *p* < 0.001), nightmares (*p* = 0.007), vivid dreams (*p* = 0.009), head explosion (*p* = 0.003), and dream-related behavior (*p* = 0.037). Violence in sleep also significantly differed based on the occurrence of neurological disorders (*p* < 0.001).

Similarly, statistically significant correlations were found between health complications in pregnancy and BDI-II (Spearman’s rho = 0.120, *p* < 0.031) and STAI-T (Spearman’s rho = 0.146, *p* < 0.009) scores after pregnancy, as well as vivid (Spearman’s rho = 0.198, *p* < 0.001) and aggressive dreams (Spearman’s rho = 0.182, *p* < 0.001), nightmares (Spearman’s rho = 0.153, *p* < 0.001) and sleep paralysis (Spearman’s rho = 0.159, *p* < 0.001) with hypnagogic hallucinations (Spearman’s rho = 0.110, *p* < 0.006) during pregnancy, but this effect did not persist after delivery.

## 4. Discussion

Our results showed a significant increase in parasomnia episodes during pregnancy in comparison to the 3 months before pregnancy and after delivery, including sleepwalking and night terrors, vivid dreams/nightmares, and head explosion. Sleep paralysis was the only parasomnia to increase after delivery. The incidence of other types of parasomnias did not significantly change during pregnancy and in the postpartum period. This is a very interesting finding compared to the existing literature. As far as we are aware, our work is only the second to compare the incidence of individual parasomnias in a longitudinal design, i.e., before pregnancy, during pregnancy, and after delivery. The first study, conducted in 2002, showed an overall decrease in the prevalence of parasomnias with a sufficient sample size, which was more pronounced in primiparas than in multiparas [11].

NREM parasomnias are characterized by abnormal nocturnal behavior, impaired consciousness, and autonomic activation due to incomplete arousal, primarily arising from NREM 3 sleep [16,19,20]. They include confusional arousal, sleepwalking, night terrors, sleep-related eating disorder, and sexsomnia, and are also the most common cause of sleep-related violence [21,22]. The incidence of NREM parasomnias in pregnancy has not been systematically studied. A single paper describes a decrease in the incidence of sleepwalking with the onset of pregnancy [11], while we found a higher incidence of both sleepwalking and night terrors. Two earlier case studies that describe women whose sleepwalking or even sleep terror episodes were exacerbated or newly appeared during pregnancy are consistent with our results [23,24]. Nevertheless, other work focusing on sleep disturbances in pregnancy compared to non-pregnant women found no significant changes in the prevalence of NREM parasomnias in pregnancy [25]. We think that our results may reflect the increased psychosocial stress associated with pregnancy as well as impaired sleep quality, and a presumed higher incidence of sleep disorders such as sleep-disordered breathing (SDB) or periodic limb movements in sleep (PLMS) [20]. We are also fully aware that without capturing the specific event on video-polysomnography, it is, in some cases, difficult to distinguish whether it is an NREM or REM parasomnia episode.

REM parasomnias are a more frequently studied group of parasomnias in the context of pregnancy, with particular attention being paid to nightmares [14], but even here the results diverge. Hedman et al. demonstrated a decrease in the frequency of nightmares through the course of pregnancy [11]. In contrast, Lara-Carrasco found an increase in masochistic content and a degree of disturbance in the dreams of pregnant women in their late third term compared to the early stages of their pregnancy [13,26]. Another study showed no significant change between the frequency of nightmares comparing pregnant women to postpartum women [12]. On the other hand, pregnant women seem to have more nightmares than healthy individuals [13,27]. These inconclusive findings may be partly due to the different sample sizes and a cross-sectional design, which is less appropriate than a longitudinal design in describing changes in the severity of parasomnias during pregnancy. Other REM parasomnias are neglected in most cases. We found a significant increase in episodes of sleep paralysis [14] in the postpartum period. Hedman et al., on the other hand, describes its maximum occurrence during the second and third trimesters of pregnancy [11]. Nielsen and Paquette investigated behavior that may be related to dreams and found that pregnant women experienced less dream-related behavior than postpartum women [12], but we did not confirm these findings.

Of the other parasomnias that do not have a distinct link to the sleep stage, we determined head explosion. The results suggest a higher incidence of this parasomnia in pregnancy; however, given the existence of many types and causes of headache in pregnancy [28], some of which may occur during the night and the inability to compare with other studies, we are cautious in our conclusions.

It appears that there is a clear impact of pregnancy on parasomnia symptoms, which may have several potential causes. One such cause may be fragmented sleep and frequent awakenings from sleep during pregnancy. This may be caused by several factors directly related to pregnancy, which need to be further investigated, e.g., using electroencephalography, endocrine hormone concentrations, or at the level of neurotransmitter changes. However, the cause of parasomnias in pregnancy is multifactorial; it is also necessary to directly take into account arousals associated with urination, or difficulty finding a comfortable position [29]. Furthermore, parasomnias may occur more frequently in association with other sleep disorders that are common in pregnancy [30,31]. Arousals secondary to hypo-/apneas and related breathing events [32], or arousal associated with periodic limb movements, were shown to precipitate episodes of NREM parasomnias [33]. Women with a medically diagnosed SDB or PLMS were not included in this study.

Furthermore, we found an association between psychiatric diagnoses and some parasomnias in all the periods studied. In addition, some studies suggest that parasomnias may be related, to a certain extent, to current psychiatric symptoms such as anxiety or depression [34]. Another study reports that levels of anxiety and depression increase during pregnancy, with the highest levels reaching their peak during the first trimester [35]. Our results, which showed a correlation between current symptoms of anxiety/depression and vivid dreams/nightmares after delivery, support these findings. Although the pathophysiological links are not yet fully elucidated, changes in neurotransmitter levels may play a role, or the inflammatory environment, which is known to be elevated in pregnant women, may be another contributing factor. Elevated inflammatory markers are also related to psychiatric problems such as depression [36]. Therefore, it is also possible that psychological changes in pregnancy, in part related to elevated inflammatory markers, may lead to a higher incidence of sleep disorders, including parasomnias. Moreover, disrupted sleep continuity also increases inflammatory markers [37], and we have shown in this text that sleep is more fragmented in pregnancy [28,29]. We did not find a convincing association with SSRI use, but this may be due to the low number of subjects taking these medications.

Another of our findings is the association of chronic headaches such as migraines with the occurrence of parasomnias in pregnancy. There are only a few studies on this topic, but they confirm the association of migraines with both dream-related behavior [38] and sleepwalking [39]. A dysfunction of the serotoninergic pathway has been hypothesized as a possible common pathological mechanism because of the well-known role of serotonin in both sleep–wake regulation and in migraine pathogenesis. The most recent clinical hypothesis regards a dysfunction of orexinergic projections on the raphe nuclei that interferes with serotonergic regulation, altering the nociceptive and the sleep regulating systems [40,41]. Other pain conditions, such as chronic tension headaches and chronic vertebral pain, may also trigger parasomnia episodes.

The limitations of the study may be its retrospective design with the possible impact of recall bias, and the lack of video-polysomnography [42,43], which may objectify the occurrence of parasomnia episodes and their association with specific sleep stages, as well as potentially determine the occurrence and severity of comorbid sleep disorders, especially those that can trigger parasomnia episodes.

## 5. Conclusions

In conclusion, our findings highlight the need to be concerned about sleep quality and sleep disturbances during pregnancy. Our results suggest that there is a need to look beyond the previously known sleep disorders that are prevalent in pregnancy, such as restless legs syndrome, sleep-disordered breathing, and insomnia. Parasomnias are also a significant problem in pregnancy. As our results show, nightmares and vivid dreams are among the most common, but they are far from the only parasomnias that need attention during pregnancy. We have observed a rise in parasomnias during pregnancy in terms of their overall incidence, but also in terms of their severity. Our results further suggest that there is a need to conduct targeted searches for coexisting neurological and psychiatric diseases and for current psychiatric status. At the same time, it should be borne in mind that medical complications in pregnancy may also be associated with a higher incidence of parasomnias, and thus, contribute significantly to poor quality sleep with all its known consequences.

Having a good understanding of the prevalence of parasomnias in pregnancy is very important given that there are now a number of non-pharmacological treatments that can significantly improve sleep quality, reduce the incidence or intensity of parasomnias, and more broadly, for example, reduce the number of injuries or accidents that may be associated with them. Of course, we cannot forget the need to treat the aforementioned sleep disorders, such as restless legs syndrome and periodic limb movements in sleep, as well as sleep-disordered breathing, the treatment of which can also significantly reduce the incidence of parasomnia episodes.

## Figures and Tables

**Figure 1 brainsci-13-00357-f001:**
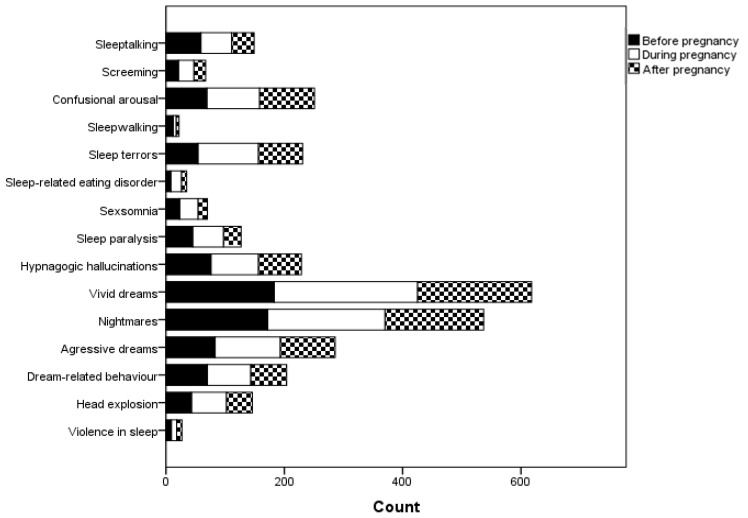
Representation of individual parasomnias in all the observed time periods.

**Table 1 brainsci-13-00357-t001:** The questions asked were related to 3 time-defined periods: (1) within 3 months before pregnancy; (2) during pregnancy; and (3) within 3 months after delivery; and the corresponding events/parasomnias.

No.	Specific Event or Behavior over 3 Time-Defined Periods	Probable Event/Parasomnia
1.	Have you ever talked in your sleep?	Sleep talking
2.	Have you ever screamed in your sleep?	Screaming
3.	Have you ever woken up disoriented or confused (with semi-purposeful movements), unresponsive to external cues?	Confusional arousal
4.	Have you ever been sleepwalking (tried to leave or left your bed asleep)?	Sleepwalking
5.	Have you ever woken up in a state of terror (with intense fear, heart pounding, sweaty)?	Sleep terror
6.	Have you ever gone from bed to eat or drink without knowing it?	Sleep-related eating disorder (SRED)
7.	Have you ever been sexually active in your sleep without knowing it?	Sexsomnia
8.	Have you ever experienced a transient state of inability to move when falling asleep or waking up?	Sleep paralysis (SP)
9.	Have you ever experienced a delusion or feeling of the presence of people, animals, things, or unpleasant events when falling asleep or waking up?	Hypnagogic hallucinations (HH)
10.	Have you ever had very vivid dreams?	Vivid dreams
11.	Have you ever had a nightmare (a scary dream that wakes you up and you clearly remember its vivid plot)?	Nightmares
12.	Did your dreams sometimes have aggressive content?	Aggressive dreams
13.	Have you ever made movements in your sleep that correspond to what you are just dreaming (i.e., waving, saluting, swatting away mosquitoes, etc.)?	Dream-related behavior
14.	Have you ever experienced a sudden, excruciating, or violent sound or headache in your sleep that woke you up?	Head explosion
15.	Have you ever (almost) hurt yourself or your bed partner in your sleep?	Violence in sleep

**Table 2 brainsci-13-00357-t002:** Frequency of parasomnia episodes in all the observed time periods.

	Before Pregnancy					During Pregnancy					After Pregnancy				
	<1/m	<1/w	1–2×/w	3–5×/w	>5×/w	<1/m	<1/w	1–2×/w	3–5×/w	>5×/w	<1/m	<1/w	1–2×/w	3–5×/w	>5×/w
Sleeptalking	266	33	17	7	2	256	41	20	4	4	272	35	13	4	1
Screaming	304	16	4	0	1	299	17	8	0	1	305	14	5	0	1
Confusional arousal	256	59	10	0	0	236	67	19	3	0	232	60	26	6	1
Sleepwalking	313	11	1	0	0	320	4	1	0	0	320	2	2	1	0
Sleep terror	271	39	13	2	0	223	72	27	3	0	250	47	19	5	4
Sleep-related eating disorder	317	5	2	1	0	307	9	7	1	1	316	4	3	1	1
Sexsomnia	302	17	6	0	0	294	21	9	1	0	309	12	3	0	1
Sleep paralysis	280	35	8	0	2	273	38	9	3	2	295	22	5	2	1
Hypnagogic hallucinations	249	57	11	7	1	245	55	15	9	1	252	43	19	10	1
Vivid dreams	142	104	51	17	11	83	104	88	27	23	132	97	65	23	8
Nightmares	153	134	27	7	4	127	116	53	22	7	158	108	40	12	7
Agressive dreams	242	63	15	5	0	215	69	28	10	3	232	71	11	10	1
Dream-related behavior	255	56	10	3	1	252	55	16	1	1	264	51	9	1	0
Head explosion	282	41	1	1	0	266	44	12	3	0	281	35	7	2	0
Violence in sleep	316	6	1	1	1	316	6	1	2	0	316	6	0	2	1

Legend: w, weekly; m, monthly.

## Data Availability

Data are available from the authors upon request.

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
