# Peer review of "Parasomnias in Pregnancy"

_brainsci, 2023, doi:10.3390/brainsci13020357_

Round 1

Reviewer 1 Report

"Parasomnias in Pregnancy" is an important paper, but it needs major revision:

1. Introduction should be extended

2. Study design isn't well described

3. Conclusion is missing

4. Language have to be improved

Author Response

Thank you for your kind consideration of the text.

1.Introduction should be extended

We believe that we do not have the possibility to extend the introduction very much. There are very few studies that directly address parasomnias in pregnancy. Extending the introduction would have to address sleep disorders in pregnancy in general, which was not the focus of this study. In an attempt to meet the reviewer's objections, we have at least added references to recent papers and meta-analyses on sleep disorders in pregnancy.

  1. Study design isn't well described

We have added a detailed description of participant recruitment.

  1. Conclusion is missing

We have added a conclusion section.

  1. Language have to be improved

Proofreading by a native speaker was performed.

Reviewer 2 Report

Review parasomnia in pregnancy

This manuscript addresses a very interesting question relating to parasomnias in pregnancy, which indirectly reflects observations of sleep stability.  It is well conceived, with three time periods selected pre-, during and and post- pregnancy and has a large outreach.  There are several critical issues that should be explained or resolved before any publication:

1.       It is mentioned that social media platforms were used.  However I see no detail in terms of who was interviewed, how they were selected and what was done with those wo did not respond at all or who submitted partial responses (2/3 of the overall responders).  These are important when assessing prevalence.  Since parasomnia was reported by 213 of the 325 who fully responded, there appears to be a high likelihood of report bias.

2.       Although the difference between during pregnancy and pre- pregnancy is statistically significant, but does not appear very impressive because it is only 25 vs 21%, is this correct?.  Also, the numbers in the second paragraph of the results are confusing – how many had parasomnias in all 3 time periods?  Is this what table 2 represents?  It may be more helpful perform the comparisons within subject and in terms of change at each time period, i.e., new parasomnia in pregnancy vs pre, and any resolution after delivery.

3.       The questionnaire lists “have you ever…” – does this mean that someone who completing the questionnaire during pregnancy  would be answering “yes” even if they had parasomnia in childhood?

4.       Vivid dreams and dream enactment can be side effects of SSRI medications – were any of these women in treatment for depression?

5.       Since there was no PSG performed, it is unknown whether these parasomnias are REM -related, NREM or even due to an abnormal arousal, please omit this part of the discussion.

6.       Similarly, no mention is given to any sleep related comorbidities, which can have near identical representation. 

7.       There should be detail given regarding how comorbid disorders were evaluated and what they were.  Of particular interest are epilepsy, PLMS, and sleep apnea. 

8.       In terms of literature, the list of cited publications should be updated with more recent publications.  There have been several recent publications in the area of parasomnia discussions that would be worth mentioning:

Gupta R, Rawat VS. Sleep and sleep disorders in pregnancy. Handb Clin Neurol. 2020;172:169-186. doi: 10.1016/B978-0-444-64240-0.00010-6. PMID: 32768087.

Chen SJ, Shi L, Bao YP, Sun YK, Lin X, Que JY, Vitiello MV, Zhou YX, Wang YQ, Lu L.

Limbekar N, Pham J, Budhiraja R, Javaheri S, Epstein LJ, Batool-Anwar S, Pavlova M. NREM Parasomnias: Retrospective Analysis of Treatment Approaches and Comorbidities. Clocks Sleep. 2022 Aug 16;4(3):374-380. doi: 10.3390/clockssleep4030031. PMID: 35997385; PMCID: PMC9397000.

Prevalence of restless legs syndrome during pregnancy: A systematic review and meta-analysis. Sleep Med Rev. 2018 Aug;40:43-54. doi: 10.1016/j.smrv.2017.10.003. Epub 2017 Oct 25. PMID: 29169861. (because this was based on questionnaire)

For evaluation:

Cesari M, Heidbreder A, St Louis EK, Sixel-Döring F, Bliwise DL, Baldelli L, Bes F, Fantini ML, Iranzo A, Knudsen-Heier S, Mayer G, McCarter S, Nepozitek J, Pavlova M, Provini F, Santamaria J, Sunwoo JS, Videnovic A, Högl B, Jennum P, Christensen JAE, Stefani A. Video-polysomnography procedures for diagnosis of rapid eye movement sleep behavior disorder (RBD) and the identification of its prodromal stages: guidelines from the International RBD Study Group. Sleep. 2022 Mar 14;45(3):zsab257. doi: 10.1093/sleep/zsab257. PMID: 34694408.

Bubrick EJ, Yazdani S, Pavlova MK. Beyond standard polysomnography: advantages and indications for use of extended 10-20 EEG montage during laboratory sleep study evaluations. Seizure. 2014 Oct;23(9):699-702. doi: 10.1016/j.seizure.2014.05.007. Epub 2014 May 24. PMID: 24939522.

Author Response

Review parasomnia in pregnancy

Thank you for your kind consideration of the text.

This manuscript addresses a very interesting question relating to parasomnias in pregnancy, which indirectly reflects observations of sleep stability.  It is well conceived, with three time periods selected pre-, during and and post- pregnancy and has a large outreach.  There are several critical issues that should be explained or resolved before any publication:

  1. It is mentioned that social media platforms were used.  However I see no detail in terms of who was interviewed, how they were selected and what was done with those wo did not respond at all or who submitted partial responses (2/3 of the overall responders).  These are important when assessing prevalence.  Since parasomnia was reported by 213 of the 325 who fully responded, there appears to be a high likelihood of report bias.

We have added a more detailed description of participant recruitment.

  1. Although the difference between during pregnancy and pre- pregnancy is statistically significant, but does not appear very impressive because it is only 25 vs 21%, is this correct?.  Also, the numbers in the second paragraph of the results are confusing – how many had parasomnias in all 3 time periods?  Is this what table 2 represents?  It may be more helpful perform the comparisons within subject and in terms of change at each time period, i.e., new parasomnia in pregnancy vs pre, and any resolution after delivery.

Yes, the difference between „during pregnancy“ and „pre-pregnancy“ is statistically significant. Although not dramatic, the difference is stable.

We have added additional explanatory information to hopefully better describe the results, and also within subject comparisons.

  1. The questionnaire lists “have you ever…” – does this mean that someone who completing the questionnaire during pregnancy  would be answering “yes” even if they had parasomnia in childhood?

The questions are related to 3 precisely defined time periods. The study is aimed at assessing changes related to the onset of pregnancy and after delivery.

In addition, no subject could have completed the questionnaire during pregnancy, for such a participant we would not have had a complete questionnaire and would have been excluded.    

  1. Vivid dreams and dream enactment can be side effects of SSRI medications – were any of these women in treatment for depression?

Yes, we have added this information to the text.

  1. Since there was no PSG performed, it is unknown whether these parasomnias are REM -related, NREM or even due to an abnormal arousal, please omit this part of the discussion.

We have modified the discussion in respect of the reviewer's request.

  1. Similarly, no mention is given to any sleep related comorbidities, which can have near identical representation. There should be detail given regarding how comorbid disorders were evaluated and what they were.  Of particular interest are epilepsy, PLMS, and sleep apnea.

       As we do not have video-polysomnography, we cannot comment on current sleep comorbidities. None  of the participants had a medically diagnosed sleep disorder. Participants' complaints were vague and did not allow us to comment on the comorbidities more closely and precisely. We also took into account the fact that the occurrence of other sleep disorders is well described in the literature. Our study was primarily focused on changes in the incidence of parasomnias with the onset of pregnancy and postpartum. It is likely that a number of parasomnias can be related to SDB or PLMS, but without video-polysomnography and episode capture it is still not possible to conclusively state that the sleep disorders are directly related. 

  1. In terms of literature, the list of cited publications should be updated with more recent publications.  There have been several recent publications in the area of parasomnia discussions that would be worth mentioning:

Gupta R, Rawat VS. Sleep and sleep disorders in pregnancy. Handb Clin Neurol. 2020;172:169-186. doi: 10.1016/B978-0-444-64240-0.00010-6. PMID: 32768087.

    – This study was between our references

Chen SJ, Shi L, Bao YP, Sun YK, Lin X, Que JY, Vitiello MV, Zhou YX, Wang YQ, Lu L. Prevalence of restless legs syndrome during pregnancy: A systematic review and meta-analysis. Sleep Med Rev. 2018 Aug;40:43-54. doi: 10.1016/j.smrv.2017.10.003. Epub 2017 Oct 25. PMID: 29169861. (because this was based on questionnaire)

Limbekar N, Pham J, Budhiraja R, Javaheri S, Epstein LJ, Batool-Anwar S, Pavlova M. NREM Parasomnias: Retrospective Analysis of Treatment Approaches and Comorbidities. Clocks Sleep. 2022 Aug 16;4(3):374-380. doi: 10.3390/clockssleep4030031. PMID: 35997385; PMCID: PMC9397000.

  • We have added these studies to the list of references.

For evaluation:

Cesari M, Heidbreder A, St Louis EK, Sixel-Döring F, Bliwise DL, Baldelli L, Bes F, Fantini ML, Iranzo A, Knudsen-Heier S, Mayer G, McCarter S, Nepozitek J, Pavlova M, Provini F, Santamaria J, Sunwoo JS, Videnovic A, Högl B, Jennum P, Christensen JAE, Stefani A. Video-polysomnography procedures for diagnosis of rapid eye movement sleep behavior disorder (RBD) and the identification of its prodromal stages: guidelines from the International RBD Study Group. Sleep. 2022 Mar 14;45(3):zsab257. doi: 10.1093/sleep/zsab257. PMID: 34694408.

Bubrick EJ, Yazdani S, Pavlova MK. Beyond standard polysomnography: advantages and indications for use of extended 10-20 EEG montage during laboratory sleep study evaluations. Seizure. 2014 Oct;23(9):699-702. doi: 10.1016/j.seizure.2014.05.007. Epub 2014 May 24. PMID: 24939522.

  • We are not sure if there is a need to add these papers as this is a questionnaire study. However, we have added them at the request of the reviewer.

Round 2

Reviewer 1 Report

"Parasomnias in Pregnancy" is an interesting paper. It improved after revision and can be published after minor revision.

- language should be checked by a native speaker

- conclusion could be extended

Author Response

Thank you very much for your comments. The language will be checked by the in-house English editors. The conclusion is extended, please see the revised manuscript.

Reviewer 2 Report

Thanks vor revising th emanuscript

Author Response

Thank you very much for your comments.